# Catheter Ablation of Frequent PVCs in Structural Heart Disease: Impact on Left Ventricular Function and Clinical Outcomes

**DOI:** 10.3390/biomedicines13061488

**Published:** 2025-06-17

**Authors:** Nikias Milaras, Nikolaos Ktenopoulos, Paschalis Karakasis, Aikaterini-Eleftheria Karanikola, Vasileios Michopoulos, Konstantinos Pamporis, Panagiotis Dourvas, Anastasios Apostolos, Zoi Sotiriou, Stefanos Archontakis, Athanasios Kordalis, Konstantinos Gatzoulis, Skevos Sideris

**Affiliations:** 1State Department of Cardiology, “Hippokration” General Hospital of Athens, 11527 Athens, Greece; panosdour92@gmail.com (P.D.); stef6arch@yahoo.com (S.A.); skevos1@otenet.gr (S.S.); 2School of Medicine, National and Kapodistrian University of Athens, Hippokration General Hospital of Athens, 11527 Athens, Greece; nikosktenop@gmail.com (N.K.); elinakaranikola@gmail.com (A.-E.K.); vmichopo@its.jnj.com (V.M.); konstantinospab@gmail.com (K.P.); anastasisapostolos@gmail.com (A.A.); akordalis@gmail.com (A.K.); kgatzoul@med.uoa.gr (K.G.); 3First Department of Cardiology, National and Kapodistrian University of Athens, Hippokration General Hospital of Athens, 11527 Athens, Greece; 4Second Department of Cardiology, Aristotle University of Thessaloniki, General Hospital “Hippokration”, 54642 Thessaloniki, Greece; pakar15@hotmail.com

**Keywords:** premature ventricular contraction, structural heart disease, ischemic cardiomyopathy, non-ischemic cardiomyopathy, ablation

## Abstract

**Background:** Frequent premature ventricular complexes (PVCs) are associated with adverse outcomes in patients with structural heart disease (SHD), including increased risk of mortality and impaired left ventricular ejection fraction (LVEF). While radiofrequency ablation (RFA) of idiopathic PVCs is well established, its role in patients with SHD remains less clear. **Objective:** To review the evidence on the efficacy of RFA for PVC suppression in patients with SHD, specifically evaluating its impact on LVEF and clinical outcomes. **Methods:** A review of the literature was conducted using PubMed and the Cochrane Library, focusing on studies published after 2010 that included adult patients with SHD and a PVC burden >4% on 24 h Holter monitoring. Studies including patients with presumed PVC-induced cardiomyopathy without underlying SHD were excluded. Key outcomes were LVEF recovery, functional status, and procedural success rates. **Results:** In ischemic cardiomyopathy, RFA reduced PVC burden significantly and resulted in modest but significant LVEF improvement. In non-ischemic cardiomyopathy, successful ablation improved LVEF by 8–12% on average and enhanced NYHA class. Across mixed cohorts, patients with sustained PVC suppression showed significant improvements in LVEF, functional status, which, in many cases, removed the indication for implantable cardioverter-defibrillators. Notably, procedural success rates ranged from 60 to 94%, and the high baseline PVC burden (>13–20%) consistently predicted LVEF recovery regardless of SHD etiology. **Conclusions:** RFA of frequent PVCs in patients with SHD leads to meaningful improvements in systolic function and symptoms, particularly in those with high PVC burden. These benefits are seen across ischemic and non-ischemic substrates, although procedural complexity and recurrence rates may be higher. PVC burden, rather than SHD presence alone, should guide patient selection for ablation.

## 1. PVC Ablation in Structural Heart Disease

A high PVC burden in the setting of structural heart disease (SHD) portends worse outcomes. The GISSI-2 trial confirmed that more than 10 PVCs per hour after an acute myocardial infarction were an independent risk factor for sudden cardiac death in the following six months [1]. This formed the central hypothesis of the CAST trial in ischemic heart disease [2]. Unfortunately, despite the effective suppression of PVCs with flecainide or encainide, there was a doubling in cardiac mortality. This unexpected result led to the understanding that simply eliminating PVCs without addressing the underlying myocardial substrate could be harmful, highlighting the proarrhythmic risks of certain antiarrhythmic drugs in vulnerable hearts. Similarly, the presence of PVCs in dilated cardiomyopathy has been linked to adverse clinical outcomes. A recent sub-study of the DANISH trial demonstrated that >30 PVCs/hour was associated with an increase in total mortality and cardiovascular death [3,4].

Frequent PVCs have also been implicated in the development of a dysfunction or worsening of contractility in cardiomyopathy, conditions now known as PVC-induced cardiomyopathy or PVC-aggravated cardiomyopathy [5]. Theories suggesting that PVCs worsen left ventricular systolic function in structural heart disease began to emerge in the early 1990s [6]. However, it was not until 1995 that the first randomized trial comparing amiodarone to placebo for PVC suppression demonstrated that suppressing PVCs often led to the recovery of LVEF, a finding that appeared to translate into a mortality benefit [7]. The radiofrequency ablation (RFA) of PVCs can lead to the effective suppression of ventricular ectopy, reversing the deleterious effects of asynchronous contraction and Ca^2+^ accumulation in the myocardium [8]. Even though numerous trials have been published in regard to the RFA of idiopathic PVCs in the absence of apparent heart disease, the evidence of RFA in individuals with systolic dysfunction of the LV remains limited. Over the past years, advancements in PVC ablation technology—including high-density electroanatomical mapping, improved catheter design, and real-time imaging integration—have significantly enhanced procedural precision and safety. These developments have increased both the feasibility and effectiveness of ablation, even in patients with structural heart disease, by enabling more accurate identification and targeted elimination of ectopic foci.

The purpose of this study was to conduct a review of the literature utilizing the PubMed and Cochrane library platforms, in order to gain insight into the usefulness of RFA in patients with SHD. In particular, we sought to unveil whether the sustained ablation of PVCs in patients with systolic dysfunction and a diagnosis of ischemic or non-ischemic cardiomyopathy led to an improvement of LVEF and functional status.

RFA in patients with myocardial scar differs significantly from RFA for idiopathic PVCs arising from the outflow tracts. To begin with, the underlying arrhythmogenic mechanisms are distinct: idiopathic PVCs are primarily driven by triggered activity or abnormal automaticity, whereas re-entry circuits around scar areas are thought to be the predominant mechanism of PVCs in SHD [4]. Interestingly, outflow-tract PVCs may occasionally be encountered even in patients with SHD [9]. Secondly, the mapping and ablation strategies may differ substantially. In idiopathic PVCs, activation mapping or pace-mapping is often sufficient to localize and eliminate the focus with relatively high success rates [9]. In contrast, in the setting of myocardial scarring, more complex substrate-based approaches are frequently required, including detailed voltage mapping to identify areas of low voltage and late potentials, which may serve as critical components of re-entrant circuits [10]. Additionally, ablation outcomes in SHD are generally less favorable compared to idiopathic cases, with higher recurrence rates and an increased risk of procedural complications. This is especially the case in patients with DCM, since the arrhythmogenic focus is frequently epicardial and therefore not amenable to endocardial RFA [11].

## 2. Methods

This study was conducted as a narrative review aimed at synthesizing current evidence regarding the role of RFA for frequent PVCs in patients with SHD. Our primary objective was to evaluate whether successful ablation led to improvement in LVEF and clinical outcomes in this population. We systematically searched PubMed and the Cochrane Library databases from inception to April 2025 using combinations of the following keywords: “PVC”, “premature ventricular contraction”, “radiofrequency ablation”, “structural heart disease”, “cardiomyopathy”, “ischemic”, “non-ischemic”, “ventricular arrhythmia”, “left ventricular dysfunction” and “LVEF improvement”. Boolean operators (“AND”, “OR”) were applied to refine the search.

Studies were eligible for inclusion if they met the following criteria:Included adult patients with documented structural heart disease (ischemic or non-ischemic);Evaluated the impact of catheter ablation on frequent PVCs (typically defined as a burden >4% on 24 h Holter monitoring);Reported at least one post-ablation outcome related to LVEF, heart failure symptoms (e.g., NYHA class), natriuretic peptide levels, or clinical endpoints;Published in English in peer-reviewed journals.

Studies focusing solely on idiopathic PVCs, pediatric populations, or those without data on structural cardiac abnormalities were excluded. Reference lists of relevant articles were manually screened to identify additional eligible studies. Given the heterogeneity in study designs and endpoints, a formal meta-analysis was not feasible. Instead, findings were summarized in tabular form and synthesized qualitatively. The risk of potential bias was considered narratively, including overlap between study cohorts, single-center designs, and lack of randomization in most studies.

## 3. Results

### 3.1. Ischemic Cardiomyopathy

Most PVC origin sites in patients after myocardial infarction are located within the infarct scar. The underlying substrate likely consists of surviving myocardial bundles within the scarred tissue. In a prospective study by Sarrazin et al., 15 patients with ischemic cardiomyopathy, low LVEF (0.36 ± 0.12) and PVC burden >5% on Holter monitoring underwent RFA and were compared to 15 controls without frequent ventricular ectopy. Mean PVC burden was reduced from 21.8 ± 12.5% to 2.6 ± 5.0%, and the procedure was considered successful in all patients. Subjects with frequent ectopy had less scarring on magnetic resonance imaging (MRI). Furthermore, the majority of patients in the RFA arm had a history of inferior myocardial infarction, suggesting that the reduced LVEF might be attributable to an alternative mechanism, such as PVCs [12]. Notably, mean LVEF increased from 0.38 + 0.11 to 0.51 + 0.09 at 6 months [13,14,15].

The same group of authors published another series of 28 patients with ischemic heart disease. In that study, catheter ablation of frequent PVCs in post-infarction patients led to a significant reduction in PVC burden and an improvement in left ventricular function. The baseline PVC burden was 15.6% ± 12.3% and decreased markedly to 2.4% ± 4.2% at follow-up (*p* < 0.001). Early post-ablation, the mean LVEF showed a small, non-significant increase from 0.37 ± 0.14 to 0.39 ± 0.13, but with longer follow-up, LVEF improved significantly to 0.43 ± 0.15 (*p* = 0.03). Acute procedural success, defined as ≥80% reduction in PVC burden, was achieved in 89% of patients, and the clinical success rate at 3–6 months was 75%. These findings highlight that ablation targeting PVCs in regions of myocardial scar can lead to meaningful clinical benefits in patients with ischemic cardiomyopathy (Table 1).

### 3.2. Non-Ischemic Cardiomyopathy

There was only one study that included patients with non-ischemic cardiomyopathy (NICMP) and that excluded those with possible PVC-induced cardiomyopathy either by the presence of myocardial scarring on cardiac magnetic resonance imaging or a clinical diagnosis predating frequent PVCs [11]. That study analyzed 30 consecutive patients (mean age 59 ± 12 years; 60% male) with NICMP and frequent PVCs (mean PVC burden 22.7% ± 11.6%) referred to for RFA. Mean baseline LVEF was 38% ± 15%. During a mean follow-up of 30 ± 28 months, successful ablation was achieved in 60% of patients, leading to a significant reduction in PVC burden (from 23.1% ± 8.8% to 1.0% ± 0.9%, *p* < 0.0001) and an improvement in LVEF (from 33.9% ± 14.5% to 45.7% ± 17%, *p* < 0.0001). NYHA functional class also improved significantly in those with successful ablation (*p* < 0.0001), while no change was observed in patients with failed procedures. In most cases, the arrhythmogenic substrate was localized within the scar tissue, typically identified via cardiac MRI. These results suggest that frequent PVCs in NICMP contribute to functional deterioration and that successful ablation can markedly improve but not always normalize cardiac function. Furthermore, the lower success rates of RFA reported in this study may be explained by the inherent complexity of non-ischemic SHD. The arrhythmogenic substrate is often diffuse and involves both endocardial and epicardial layers, making effective ablation more challenging since typical endocardial approaches often fail. Additionally, scarring is frequently located on the epicardial surface, which limits the efficacy of endocardial ablation strategies and may necessitate more advanced mapping and access techniques (Table 2).

### 3.3. Mixed Etiologies

In an intriguing study by Penela et al., 66 patients (mean age 53 ± 13 years, 50% male) with severe left ventricular dysfunction (mean LVEF 28% ± 4%) and frequent PVCs (mean PVC burden 21% ± 12%) who met criteria for a primary prevention implantable cardioverter-defibrillator (ICD) were prospectively included [13]. Of these, 17% had ischemic heart disease, while the remainder were classified as NICMP after the exclusion of epicardial coronary artery disease. After undergoing PVC ablation, patients showed significant clinical and functional improvement, with LVEF increasing to 42% ± 12% at 12 months (*p* < 0.001) and NYHA class improving markedly. Sustained ablation success, defined as a persistent ≥80% PVC burden reduction, was achieved in 76% of the cohort. Notably, the ICD indication was removed in 64% of patients during follow-up, particularly in those with a baseline PVC burden ≥13% and sustained ablation success. Throughout the study period, there were no sudden cardiac deaths or malignant ventricular arrhythmias. Based on the LVEF increase after successful ablation, 28% of patients were classified as non-ischemic cardiomyopathy at 12 months since the LVEF improved but remained <50%, and 26% were categorized as PVC-induced cardiomyopathy since LVEF normalized (Table 3).

Another prospective multicenter study by the same group enrolled 80 consecutive patients (mean age 53 ± 12 years; 59% male) with left ventricular dysfunction (mean LVEF 34.3% ± 13%) and frequent premature ventricular complexes (mean burden 22% ± 13%) [14]. Among them, 34% had a previously diagnosed SHD (besides non-ischemic cardiomyopathy), including 17 patients with ischemic heart disease, 4 with noncompaction cardiomyopathy, and 2 with valvular heart disease. The majority of patients underwent cardiac MRI imaging with LGE being present in only 19%. Following RFA, successful ablation was achieved in 66% of patients, leading to significant improvements in LVEF (from 33.7% ± 8% to 45.8% ± 10.9% at 12 months; *p* < 0.05) and reductions in BNP levels. NYHA functional class also improved markedly, with the proportion of patients in class I increasing from 23% to 79% at 12 months (*p* < 0.05). Importantly, the degree of improvement was independent of SHD status and strongly associated with baseline PVC burden and the persistence of ablation success. This is another study where a baseline PVC burden ≥13% predicted LVEF recovery after successful RFA. There was no difference in echocardiographic response between patients with known SHD and the rest of the cohort. In patients with successful RFA and a baseline PVC burden ≥13%, no indication for primary prevention ICD implantation remained at 6 months post-ablation. These findings suggest that in patients with LV dysfunction, the evaluation and ablation of frequent PVCs can yield substantial clinical and functional recovery, irrespective of underlying structural abnormalities.

The last relevant study by this group, which was also a prospective multicenter study that included 101 consecutive patients (mean age 56 ± 12 years; 61% male) with left ventricular systolic dysfunction (mean LVEF 32% ± 8%) and frequent PVCs (mean burden 21% ± 12%) who underwent RFA, was the only available study in the literature reporting clinical outcomes such as cardiac mortality and heart failure hospitalization [15]. Approximately 35% had previously diagnosed structural heart disease, including ischemic (63%) and non-ischemic etiologies. Over a mean follow-up of 34 ± 16 months, ablation was acutely successful in 94% of patients and led to a significant and sustained reduction in PVC burden (to 3.8% ± 6%) and a corresponding improvement in LVEF (39  ±  12%), NYHA functional class, and brain natriuretic peptide levels (all *p* < 0.001). Most of the benefits occurred within the first 6 months post-ablation and persisted during long-term follow-up. A sustained reduction of at least 18 points in baseline PVC burden emerged as an independent predictor of improved survival, being significantly associated with a lower risk of cardiac mortality, cardiac transplantation, or hospitalization for heart failure. These findings support the prognostic importance of durable PVC suppression following ablation in patients with LV systolic dysfunction, regardless of underlying structural abnormalities.

In a French retrospective cohort study, 168 patients with presumed PVC-induced cardiomyopathy were categorized in a low LVEF = 38% ± 10 group and a group with normal LVEF [16]. Both groups underwent RFA. There was a subgroup of patients with SHD (n = 37, of whom ischemic heart disease = 22, NICMP = 8), comprising 39% of those diagnosed with possible PVC-induced cardiomyopathy, in whom the successful RFA of PVCs was associated with significant improvements in cardiac function. Among the 25 patients with SHD who achieved a sustained reduction in PVC burden (≥80%), LVEF improved from 34% ± 9% to 49% ± 11% (*p* < 0.0001), and left ventricular end-diastolic diameter decreased from 65 ± 9 mm to 60 ± 7 mm (*p* = 0.01). In contrast, the 12 SHD patients in whom ablation was unsuccessful showed no meaningful changes in LVEF (33% ± 15% to 34% ± 16%) or left ventricular end-diastolic diameter (68 ± 13 mm to 65 ± 13 mm).

In a prospective single-center study, Arafa et al. assessed the efficacy of RFA in 77 patients with impaired LVEF and a >10%/24 h burden of PVCs, with or without known SHD [17]. Approximately 56% of the cohort had underlying SHD, including 40 with ischemic heart disease, 14 with dilated cardiomyopathy, and 7 with significant valvular disease. The mean baseline LVEF in the SHD group was 36.8 ± 7.1, the average PVC burden was 30.76 ± 9.91, and the left ventricular end-diastolic diameter was 61.4 ± 6.9. Despite a higher procedural complexity in those patients—as evidenced by longer fluoroscopy time and more ablation trials—acute and long-term procedural success rates were high and comparable across groups (initial success: ~92%; long-term success: ~81%). Of note, the outflow tract was the PVC location in 78% of the SHD group. At six months post-ablation, LVEF significantly improved to 47.2 ± 11.8, with 75% of successfully ablated patients showing >5% LVEF improvement. In a similar way, left ventricular end-systolic and end-diastolic dimensions improved. Importantly, the magnitude of LVEF recovery did not differ significantly between those with and without SHD, although ischemic cardiomyopathy cases exhibited the least improvement. A multivariate analysis identified PVC burden before and after ablation as the only independent predictors of LVEF recovery, with optimal thresholds of >18% pre-ablation and <8% post-ablation demonstrating high sensitivity and specificity.

In the last study included in this review, 65 patients with SHD and mildly reduced LVEF (59.6% of the total cohort, 29 NICMP, 17 ischemic cardiomyopathy) had a higher baseline PVC burden (22,267 ± 12,934/day) compared to those without SHD (*p* = 0.005). Ablation was successful in 83.1% of SHD patients, leading to a mean LVEF increase from 45.2% ± 14.3% to 50.9% ± 13.5% (*p* < 0.01). Notably, the magnitude of LVEF improvement in SHD patients (5.7% ± 1.37%) was comparable to that observed in non-SHD patients, suggesting that arrhythmia suppression could yield meaningful functional recovery even in structurally abnormal hearts. A multivariate analysis further confirmed that a PVC burden >20,000/day was the only independent predictor of LVEF improvement (OR 3.53, *p* = 0.023), while the presence of SHD did not predict response. These findings underscore the clinical benefit of PVC ablation in SHD patients and suggest that a high PVC burden, rather than structural substrate alone, should guide intervention decisions.

**Table 3 biomedicines-13-01488-t003:** Studies in patients with mixed etiologies of cardiomyopathy.

Author	No. of Patients and Patient % on Antiarrhythmic Drugs	Success Rates (at Least 80% Burden Reduction)	PVC Burden Pre/Post-Ablation	LVEF Pre/Post-Ablation	LVEDD Pre/Post-Ablation	BNP Pre/Post-Ablation	NYHA Class Pre/Post-Ablation	Cardiac Mortality Reduction
Penela 2013 [14]	n = 80 (17 ICMP, 4 non-compaction, 2 valvular heart disease) b-blocker 85%, amiodarone 19%	85% acutely, 66% in 12 months	22 ± 13% to not declared	33.7 ± 8% to 45.8 ± 10.9%	59.5 ± 5.9 mm to 54.9 ± 6.1 mm in successful procedures	109 to 60 pg/mL in successful procedures	From 12 patients (23%) with NYHA I at baseline to 42 (79%)	-
Penela 2015 [13]	n = 66 (11 ICMP, 3 non-compaction, 1 valvular heart disease, the rest NICMP)	94% acutely to 76% at 6 months	21 ± 12 to not declared, in all patients	28% ± 4% to 42% ± 12% in all patients	61 ± 6 to 57 ± 6 in all patients	246 ± 187 to 176 ± 380 pg/mL in all patients	2 patients with NYHA I (3%) at baseline to 35 (53%) in all patients	-
Blaye-Felice 2016 [16]	n = 96 (22 ICMP, 4 valvular heart disease, 1 myocarditis)	79% in 24 months	26 ± 12 to 4 ± 7% in successful procedures	38 ± 10 to 50 ± 13% in all patients	62 ± 8 to 57 ± 8 in all patients	-	NYHA class I from 49% to 83% in all patients	
Wojdyła-Hordyńska 2017 [18]	n = 65 (29 NICMP, 17 ICMP, 7 valvular heart disease)	83.1% at 6 months	22,267 ± 12,934 to 2172 ± 3692 in all patients with SHD	45.2% ± 14.3% to 50.9% ± 13.5% in all patients with SHD	56.1 ± 8.4 to not declared	-	-	-
Abdelhamid 2018 [17]	n = 42 (18 ICMP, 14 NICMP, 7 valvular disease) b-blocker 93%, amiodarone 55%	90.4% acutely, 78.5% at 6 months in SHD	30.76 ± 9.91 to 4.8 ± 11.45 in all patients with SHD	36.8 ± 7.1 to 47.2 ± 11.8 in all patients with SHD	61.4 ± 6.9 to 57.4 ± 5.9 in all patients with SHD	-	-	-

Abbreviations NICMP: non-ischemic cardiomyopathy, ICMP: ischemic cardiomyopathy, PVC: premature ventricular contraction, SHD: structural heart disease, LVEDD: left ventricular end-diastolic diameter, BNP: brain natriuretic peptide, NYHA: New York Heart Association.

## 4. Discussion

This review highlights the evolving role of PVC ablation in patients with structural heart disease, a group historically underrepresented in clinical trials. The degradation of systolic function in patients with a previous diagnosis of cardiomyopathy with a frequent PVC count should always raise the suspicion of PVC-aggravated cardiomyopathy. A growing body of literature now suggests that successful abolition of the arrhythmia usually leads to an improvement in the LVEF by 10 to 15%, irrespective of the underlying cardiac pathology (Figure 1). Furthermore, an improvement in the patient’s functional status is often anticipated, such as a change in NYHA functional class and a decrease in left ventricular end-diastolic dimension and neurohormonal markers (NT-pro BNP and BNP).

While patients with ischemic cardiomyopathy may exhibit modest gains in LVEF, likely due to fixed scar burden, those with non-ischemic substrates often experience more pronounced recovery, provided the ablation is successful. Importantly, many studies have demonstrated that the presence of SHD alone does not preclude a favorable response to ablation. Rather, PVC burden consistently emerged as the dominant predictor of post-ablation improvement, with thresholds >13–20% identifying patients most likely to benefit.

In ischemic heart disease, studies by Sarrazin et al. showed that frequent PVCs, particularly in the setting of preserved myocardial viability, could significantly impair systolic function and that a successful RFA of these ectopic foci led to meaningful recovery in LVEF [12]. These findings are particularly notable given the historical assumption that systolic dysfunction in post-infarction patients is solely attributable to irreversible scar. The presence of a lower-than-anticipated LVEF, e.g., 35% in a previous inferior myocardial infarction with a high PVC burden, likely represents a reversible contributor to dysfunction [19].

Outcomes in NICMP, while generally favorable, are more heterogeneous and technically challenging. This is attributed to the fact that the arrhythmogenic substrate often involves the mid-myocardium and/or epicardium. Such scar distribution limits the effectiveness of standard endocardial ablation approaches, often necessitating more advanced mapping techniques or epicardial access. Furthermore, this patient population is highly heterogenous and many subjects with PVC-induced cardiomyopathy might have been categorized as dilated or non-ischemic cardiomyopathy in the literature presented. This is a significant limitation, since robust diagnostic criteria are not currently available. Nevertheless, in most of the included studies, patients were classified as having NICMP if the diagnosis of SHD preceded the detection of a high PVC burden. The complexity of the arrhythmic substrate and the need for advanced imaging and mapping strategies, including potential epicardial access, must be acknowledged when considering ablation in these patients.

Of note, a long-term outcome study demonstrated that a durable RFA with reduction in PVC count resulted in a more favorable prognosis such as reduced cardiac mortality, heart failure hospitalization, and transplant-free survival [15]. This finding underscores the prognostic relevance of PVC suppression in SHD, regardless of etiology. It also strengthens the rationale for incorporating PVC burden as a modifiable target in the management of patients with LV dysfunction.

Importantly, the effectiveness of ablation appears to hinge on an adequate PVC burden reduction. Across multiple prospective studies, sustained suppression of PVCs—typically defined as a ≥80% burden reduction—was associated with significant improvements in LVEF, NYHA functional class, and natriuretic peptide levels. Procedural success rates varied based on substrate complexity. Endocardial ablation was often sufficient in ischemic cardiomyopathy, whereas non-ischemic cardiomyopathy frequently necessitated advanced mapping or epicardial access due to diffuse or intramural scars. In such cases, limited accessibility of the substrate remains a key challenge, reflected in lower acute success rates and higher recurrence. Epicardial access also increases the risk of complications. RFA, however, is generally associated with a low overall complication rate (2.9%), primarily related to pericardial puncture [20]. Nevertheless, medical therapy always has a place in PVC suppression, even though there are not many options in this group of patients with an abnormal myocardial substrate.

One of the key limitations in current clinical practice is the absence of standardized criteria for selecting patients with SHD and frequent PVCs for ablation. Existing studies often use variable thresholds for PVC burden, LVEF, and symptom severity, making it difficult to generalize findings. Moreover, other important factors—such as myocardial scar distribution, response to medical therapy, and arrhythmia morphology—are inconsistently considered. Future efforts should aim to develop unified, evidence-based selection criteria that integrate imaging, electrophysiological, and clinical markers to better identify patients who are most likely to benefit from ablation.

## 5. Limitations

Despite these encouraging results, several limitations should be acknowledged. Most available studies were relatively small, non-randomized, and conducted at high-volume centers with substantial experience in mapping and ablation. As many of the studies included in this review involved overlapping research teams, it is likely that there was also an overlap in the patient populations. Patient selection likely plays a critical role in the success of these procedures, and standardized criteria for identifying candidates who will benefit most from PVC ablation are still lacking. Additionally, the role of cardiac imaging, particularly cardiac MRI with late gadolinium enhancement, which was utilized in some studies, may characterize the substrate more accurately and guide procedural planning.

## 6. Conclusions

In summary, frequent PVCs are not merely epiphenomena in structural heart disease but may actively contribute to the progression of LV dysfunction. Catheter ablation offers a viable therapeutic strategy for PVC suppression and is associated with significant improvements in cardiac function, symptom burden, and potentially prognosis. These benefits appear consistent across both ischemic and non-ischemic etiologies, although procedural success is generally higher in ischemic patients. Future large-scale, randomized studies are needed to confirm these findings, define optimal selection criteria, and integrate PVC ablation into broader heart failure management algorithms.

## Figures and Tables

**Figure 1 biomedicines-13-01488-f001:**
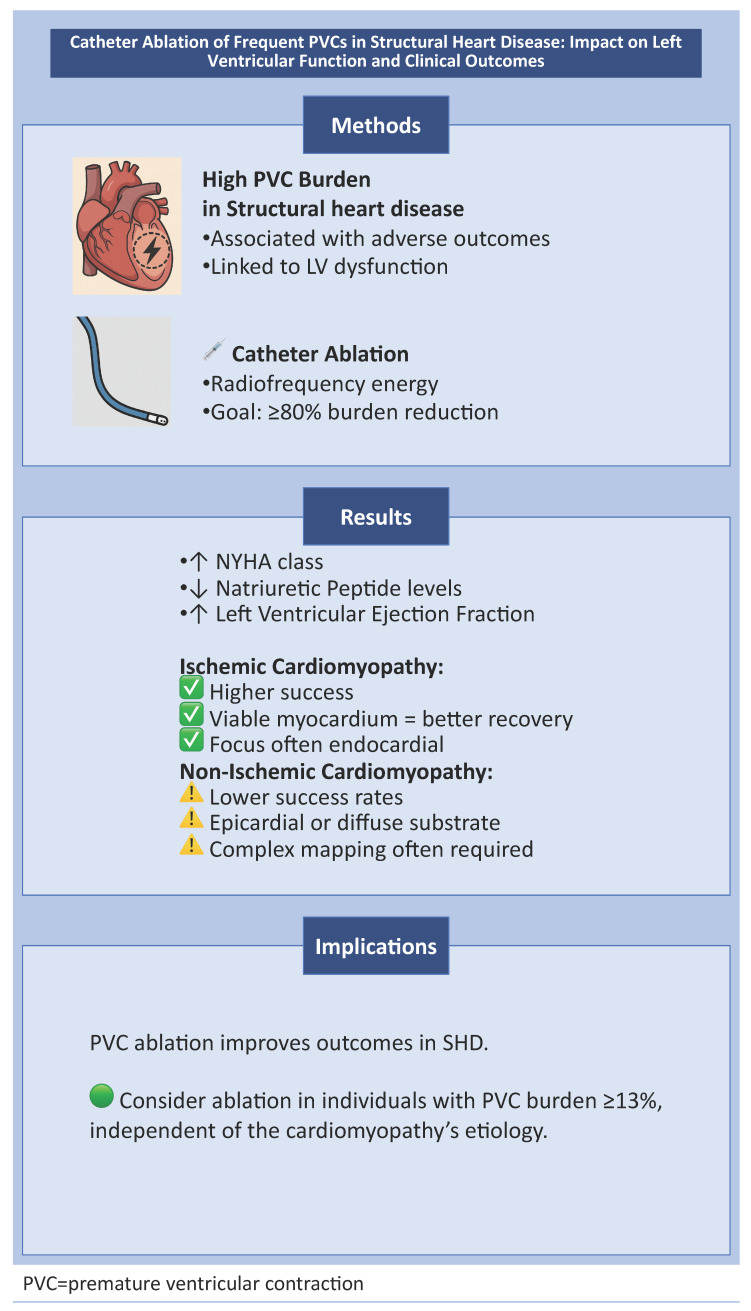
This is a central illustration summarizing the effects of catheter ablation of frequent premature ventricular contractions (PVCs) in structural heart disease (SHD). Successful ablation improves left ventricular ejection fraction and clinical outcomes, particularly in patients with high PVC burden (>13–20%), regardless of underlying SHD etiology.

**Table 1 biomedicines-13-01488-t001:** Studies in patients with ischemic cardiomyopathy.

Author	No. of Patients and Patient % on Antiarrhythmic Drugs	Patients	Success Rates (at Least 80% Burden Reduction)	PVC Burden Pre/Post-Ablation	LVEF Pre/Post-Ablation	LVEDD Pre/Post-Ablation	BNP Pre/Post-Ablation	NYHA Class Pre/Post-Ablation	Cardiac Mortality Reduction
Sarrazin 2009 [12]	n = 30 All ICMP (15 ablated, 15 controls) b-blocker 67%, amiodarone 8%, Sotalol 4%	n = 30 All ICMP (15 ablated, 15 controls)	100% at 3 to 6 months	21.8 ± 12.5 to 2.6 ± 5.0% all patients	0.38 ± 0.11 to 0.51 ± 0.09 all patients	56 ± 11 mm vs. 51 ± 8 mm all patients	-	1.8 ± 0.8 to 1.3 ± 0.5 all patients	-
Berruezo 2019 [15]	n = 101 (22 ICMP)	n = 101 (22 ICMP)	94% acutely	21 ± 12% to 3.8 ± 6% in all patients	32 ± 8% to 39 ± 12% in all patients	62 ± 7 mm to 59 ± 6 mm in all patients	36 (78–321) to 68 (32–144) pg/mL in all patients	2.2 ± 0.6% at baseline to 1.3 ± 0.6% in all patients	HR = 0.18 (95% CI: 0.05–0.66; *p* = 0.01) in successful procedures

Abbreviations ICMP: ischemic cardiomyopathy, PVC: premature ventricular contraction, SHD: structural heart disease, LVEDD: left ventricular end-diastolic diameter, BNP: brain natriuretic peptide, NYHA: New York Heart Association.

**Table 2 biomedicines-13-01488-t002:** Studies in patients with non-ischemic cardiomyopathy.

Author	No. of patients and PATIENT % on Antiarrhythmic Drugs	Patients	Success Rates (at Least 80% Burden Reduction)	PVC Burden Pre/Post-Ablation	LVEF Pre/Post-Ablation	LVEDD Pre/Post-Ablation	BNP Pre/Post-Ablation	NYHA Class Pre/Post-Ablation	Cardiac Mortality Reduction
El Kadri 2015 [11]	n = 30 All NICMP b-blocker 63% amiodarone 3%, Sotalol 3%, Dofetilide 4%	n = 30 All NICMP	60% at 48 months	23.1% ± 8.8% to 1.0% ± 0.9% in successful procedures	33.9% ± 14.5% to 45.7% ± 17% in successful procedures	58.8 ± 9.5 to 56.11 ± 8.9 in all patients	-	2.3 ± 0.6 to 1.1 ± 0.2 in successful procedures	-

Abbreviations NICMP: non-ischemic cardiomyopathy, PVC: premature ventricular contraction, SHD: structural heart disease, LVEDD: left ventricular end-diastolic diameter, BNP: brain natriuretic peptide, NYHA: New York Heart Association.

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
