# Peer review of "Catheter Ablation of Frequent PVCs in Structural Heart Disease: Impact on Left Ventricular Function and Clinical Outcomes"

_biomedicines, 2025, doi:10.3390/biomedicines13061488_

Round 1

Reviewer 1 Report

Comments and Suggestions for Authors

the authors addressed an important issue, namely the RFA of >13-20% PVC burden in structural heart disease. However, most of the patients with SDH already take beta blokers and RAS blockers, which have favorable effects in reducing the PVC burden. What do the authors consider about this issue? What are the medication doses in the studies the authors mentioned in the Table 1? I advise adding medications and doses in Table 1 for a better comprehension. Moreover, the authors had better discuss this issue in the text.

English of the last sentence of the introduction paragraph should be improved: '' amenable'' ------'' not amenable''

Table 1: study by El Kadri. the PVC burden and the LVEF increase data were written on the wrong collumns.

Author Response

We thank the reviewer for this insightful comment. We fully agree that guideline-directed medical therapy, particularly beta-blockers and renin-angiotensin system  inhibitors, play an important role in suppressing premature ventricular contractions. 

Additionally, in response to the reviewer’s suggestion, we have revised Tables to include available data on background medical therapy  as reported in the original studies. In all studies, however, detailed dosing data were not provided. 

Reviewer Comment:

English of the last sentence of the introduction paragraph should be improved: ‘‘amenable’’ ——’‘not amenable’’

Author Response:

Thank you for pointing this out. We have corrected the sentence in the introduction .

Reviewer Comment:

Table 1: study by El Kadri. The PVC burden and the LVEF increase data were written on the wrong columns.

Author Response:

We appreciate the reviewer’s careful reading. This error has now been corrected in Table 1. The PVC burden and the change in LVEF for the El Kadri et al. study are now properly aligned under their respective columns.

Reviewer 2 Report

Comments and Suggestions for Authors

General review: This manuscript offers a comprehensive review of the current literature on the efficacy of radiofrequency ablation (RFA) for frequent premature ventricular complexes (PVCs) in patients with structural heart disease (SHD). The manuscript is generally well-written and the information is presented logically. The inclusion of a graphical abstract is a commendable effort to summarize the key findings.

Specific review: 

  1. Introduction: please briefly describe the evolution of mapping and ablation technologies that have made these procedures more feasible in complex SHD substrates.
  2. Results:
  1. Please elaborate on why the substrate in NICMP is often more complex (e.g., patchy fibrosis, intramural circuits).
  2. Table 1: please consider adding a column for "Key Finding(s)" or "Main Conclusion" for each study to make the table even more informative at a glance. Also ensure all abbreviations are defined directly in the table caption or legend if not universally understood.
  3. Figure 1: for "Non-Ischemic Cardiomyopathy," the upward arrow for "Lower success rates" is confusing; perhaps use a down arrow or rephrase.

      3. Discussion: 

  1. Please consider briefly touch upon the role of antiarrhythmic drugs as an alternative or adjunct to ablation in this population, and why ablation might be preferred in certain scenarios (e.g., drug intolerance/inefficacy, proarrhythmia risk).
  2. The point about lower LVEF than anticipated in inferior MI with high PVC burden representing a reversible component is important and could be slightly expanded.
  3. While explaining the limitation, expand slightly on how standardized criteria for patient selection are lacking and what factors might need to be incorporated into future criteria.

Author Response

We would like to thank the reviewer for their thoughtful and constructive comments. We have carefully addressed each point as outlined below:

Comment: “Please briefly describe the evolution of mapping and ablation technologies that have made these procedures more feasible in complex SHD substrates.”

Response: Thank you for the suggestion. We have added a brief description in the introduction highlighting the technological advancements such as high-density mapping systems, improved catheter maneuverability, and integration of imaging modalities. (66-72).

Comment: “Please elaborate on why the substrate in NICMP is often more complex (e.g., patchy fibrosis, intramural circuits).”

Response: We agree and have expanded the discussion section to note that NICMP often presents with heterogeneous and patchy myocardial fibrosis, mid-wall scarring, and intramural arrhythmogenic foci (300-304) and (323-329).

Comment: “Please consider adding a column for ‘Key Finding(s)’ or ‘Main Conclusion’ for each study to make the table even more informative at a glance.”

Response: 

We appreciate the reviewer’s suggestion to include a “Key Finding(s)” column in Table 1. However, we chose to keep the table focused on methodological and outcome parameters to preserve clarity and avoid overcrowding. The key findings of each study are already summarized in the main text, and we believe this format ensures better readability without redundancy. We have split table 1 into 3 tables according to cardiomyopathy etiology in order to make it visually appealing.

Comment: “Ensure all abbreviations are defined directly in the table caption or legend if not universally understood.”

Response: All abbreviations in Tables have been defined below the tables.

Figure 1:

Comment: “The upward arrow for ‘Lower success rates’ is confusing; perhaps use a down arrow or rephrase.”

Response: Thank you for this observation but this is intented to be an exclamation mark within a triangle that suggests "Caution".

Comment: “Please consider briefly touching upon the role of antiarrhythmic drugs as an alternative or adjunct to ablation…”

Response: We have added a paragraph in the discussion outlining the role of antiarrhythmic drugs  (327-329). Furthermore we added the percentage of antiarrhythmic drugs give to patients in the included trial (where possible).

Comment: “The point about lower LVEF than anticipated in inferior MI with high PVC burden representing a reversible component is important…”

Response: Thank you for this comment.

Comment: “Expand slightly on how standardized criteria for patient selection are lacking…”

Response: As suggested, we have elaborated on this limitation in the discussion (330-337).

Reviewer 3 Report

Comments and Suggestions for Authors

This review article presents a comprehensive synthesis of current evidence on the efficacy and outcomes of catheter ablation  of frequent premature ventricular contractions (PVCs) in patients with structural heart disease (SHD). The authors aim to bridge a knowledge gap regarding the role of RFA in a complex and often underrepresented population: those with SHD and concurrent frequent PVCs.

The manuscript is clinically relevant, timely, and well-structured. The inclusion of both ischemic and non-ischemic subgroups, as well as the emphasis on functional outcomes such as NYHA class and LVEF improvement, enhances the translational value of the article. However, I have some recomandations:

  1. A introduction should be added, regarding AF, catether ablation and PVCs. Please see also: PMID: 37374152, PMID: 39201063.
  2. The methodology for study selection lacks sufficient detail. It is unclear how studies were screened, what inclusion/exclusion criteria were specifically applied (beyond PVC burden >4% and exclusion of idiopathic cases), and whether any quality assessment of studies was performed. A PRISMA-style approach or at least a flow diagram would strengthen this section.
  3. Several studies cited stem from overlapping cohorts (e.g., Penela et al., Berruezo et al.). While this is acknowledged, the risk of over-representation of certain findings should be further discussed or addressed with a sensitivity analysis or stratified summary.
  4. The review emphasizes improvement in LVEF, NYHA class, and natriuretic peptide levels, but does not sufficiently explore potential harms, procedural complications, or recurrence rates, particularly in NICMP patients where epicardial ablation may be required.
  5. The results section could benefit from a meta-analytic summary (if appropriate) or at least graphical presentation (e.g., forest plot or bar chart) to visually show LVEF improvements and success rates across studies.

Author Response

Reviewer Comment: An introduction should be added regarding AF, catheter ablation, and PVCs. Please see also: PMID: 37374152, PMID: 39201063.

Response: Although we thank the reviewer for his comment, we did not find the cited articles relevant to the aims and scope of our manuscript.

Reviewer Comment: The methodology for study selection lacks sufficient detail. It is unclear how studies were screened, what inclusion/exclusion criteria were specifically applied (beyond PVC burden >4% and exclusion of idiopathic cases), and whether any quality assessment of studies was performed. A PRISMA-style approach or at least a flow diagram would strengthen this section.

Response: We included a methods section addressing the issues raised..

Reviewer Comment: Several studies cited stem from overlapping cohorts (e.g., Penela et al., Berruezo et al.). While this is acknowledged, the risk of over-representation of certain findings should be further discussed or addressed with a sensitivity analysis or stratified summary.

Response: Thank you for highlighting this concern. While this is very important, it overcomes the aims of a narrative review. We acknowledged this in the discussion section, therefore prompting readers to carefully and critically appraise our findings.

Reviewer Comment: The review emphasizes improvement in LVEF, NYHA class, and natriuretic peptide levels, but does not sufficiently explore potential harms, procedural complications, or recurrence rates, particularly in NICMP patients where epicardial ablation may be required.

Response: We agree with the reviewer that a balanced discussion is essential. Procedural complications were not available in most of the included studies. We include in Tables (where available) acute and long term procedural success rates.

Reviewer Comment: The results section could benefit from a meta-analytic summary (if appropriate) or at least graphical presentation (e.g., forest plot or bar chart) to visually show LVEF improvements and success rates across studies.

Response: We appreciate this valuable suggestion. A formal meta-analysis style Forest plot was not feasible due to heterogeneity in study design, endpoints, and reporting. Some studies report efficacy only in patients with successful ablation while others report efficacy in the whole group.

Reviewer 4 Report

Comments and Suggestions for Authors

Congratulations to the authors for having produced such an interesting manuscript; I’ve found just a few weak sides that should be corrected:

In the “Methods” section it would be appropriate to specify the exact search strategies or search terms used in PubMed and Cochrane; if any additional inclusion or exclusion criteria for studies, beyond publication year and SHD status. Whether this review was conducted as a systematic or narrative review, and how bias in the included studies was assessed (even if only narratively).

Given the high complexity of patients with structural heart disease (SHD), the authors should include within their discussion a brief paragraph on the potential use of mechanical circulatory support systems during ablation in the most fragile patients (doi: 10.3390/jcm13061746.)

Although the review comprehensively discusses ablation in ischemic and non-ischemic cardiomyopathy, further expansion are needed, such as providing a clearer rationale for patient selection for ablation (for example, the role of cardiac MRI LGE in identifying patients most likely to benefit from RFA) and briefly mentioning advances in AI-driven mapping and imaging technologies that could enhance outcomes in these complex SHD cases.

Additionally, the discussion section refers to the challenges of epicardial access ( but does not mention the actual use of epicardial mapping or ablation as a practical solution

Lastly, Table 1 is pretty dense and challenging to read. Consider splitting it into two separate tables—one for ischemic cases and one for non-ischemic cases—or adding subheadings to improve readability.

Author Response

Reviewer Comment 1: In the “Methods” section it would be appropriate to specify the exact search strategies or search terms used in PubMed and Cochrane; if any additional inclusion or exclusion criteria for studies, beyond publication year and SHD status. Whether this review was conducted as a systematic or narrative review, and how bias in the included studies was assessed (even if only narratively).

Response: We thank the reviewer for this important suggestion. We have now added a “Methods” section to clarify that this review was conducted as a structured narrative review. The specific search terms used in PubMed and Cochrane have been added, and we have added inclusion and exclusion criteria beyond structural heart disease and publication year. Furthermore, we have included a brief description of our approach to assessing the quality and potential bias of included studies, performed narratively due to the heterogeneous nature of the evidence.

Reviewer Comment 2: Given the high complexity of patients with structural heart disease (SHD), the authors should include within their discussion a brief paragraph on the potential use of mechanical circulatory support systems during ablation in the most fragile patients (doi: 10.3390/jcm13061746).

Response: We appreciate this insightful recommendation. Use of mechanical circulatory support is generally not utilized nor recommended during PVC ablation. It may be recommended in VT ablation as in the article cited. Therefore we think this is beyond the scope of our review.

Reviewer Comment 3: Although the review comprehensively discusses ablation in ischemic and non-ischemic cardiomyopathy, further expansion are needed, such as providing a clearer rationale for patient selection for ablation (for example, the role of cardiac MRI LGE in identifying patients most likely to benefit from RFA) and briefly mentioning advances in AI-driven mapping and imaging technologies that could enhance outcomes in these complex SHD cases.

Response: Thank you for this constructive comment. We added a paragraph in the discussion where we describe that standarized criteria for PVC ablation do not exist. Furthermore, we address that LGE presence on MRI did not predict ablation results.

Reviewer Comment 4: Additionally, the discussion section refers to the challenges of epicardial access but does not mention the actual use of epicardial mapping or ablation as a practical solution.

Response: We agree with the reviewer and have now changed the discussion to explicitly acknowledge epicardial mapping and ablation as practical and often necessary strategies, especially in NICMP patients with epicardial or intramural substrate. (310-312 and 323-329)

Reviewer Comment 5: Lastly, Table 1 is pretty dense and challenging to read. Consider splitting it into two separate tables—one for ischemic cases and one for non-ischemic cases—or adding subheadings to improve readability.

Response: We appreciate the reviewer’s feedback regarding table clarity. We have opted to use the single-table format for better side-by-side comparison. Anyway, we split the table into 3 Tables according to caridomyopathy aetiology, as is the results section.

Round 2

Reviewer 1 Report

Comments and Suggestions for Authors

In table 2, el Kadri et.al

Please correct the minor error. The informations regarding the pvc burden decrease and LVEF increase following the ablation are in the wrong collumns. Simply place these informations to the correct collumn

Author Response

After the major revision this one seems to have gotten away. Thank you

Reviewer 3 Report

Comments and Suggestions for Authors

After reviewing the revised version of the manuscript entitled “Catheter Ablation of Frequent PVCs in Structural Heart Disease: Impact on Left Ventricular Function and Clinical Outcomes”, I acknowledge that the authors have addressed several of the earlier comments. However, I believe the manuscript still requires major revisions before it can be reconsidered for publication. The following key issues remain:

  1. Title and Scope Discrepancy

The current title suggests a comprehensive overview of PVC ablation in the context of structural heart disease. However, no mention is made of atrial fibrillation (AF) in the introduction or main text, despite its clinical relevance and frequent coexistence in this patient population. Moreover, AF-related mechanisms and ablation strategies are entirely omitted. If AF is outside the intended scope, the title should be revised accordingly to reflect the true focus.

  1. Insufficient Contextual Introduction

The introduction lacks adequate background on AF, PVC mechanisms, and catheter ablation in SHD, which are essential for establishing the rationale and clinical relevance of the review. A more comprehensive and better referenced introductory section would enhance the scientific grounding of the review.

  1. Unmarked Revisions

The authors did not indicate which parts of the manuscript were revised. This significantly hinders the review process, as changes cannot be easily identified or assessed in context. All changes should be clearly marked (e.g., tracked or highlighted) in the resubmission to facilitate proper re-evaluation.

  1. Superficial Treatment of Procedural Complications

While the benefits of PVC ablation are well covered (improvements in LVEF, NYHA class, BNP), the risks, complications, and recurrence rates remain underdeveloped, particularly in non-ischemic populations where epicardial access is more common. A balanced view is critical for clinical applicability.

  1. No Graphical Summary or Visual Data Synthesis

Although the authors argue that meta-analysis is not feasible due to heterogeneity, the manuscript would greatly benefit from visual presentation of key findings (e.g., bar charts showing LVEF changes or success rates across studies). This would improve clarity and reader engagement.

Author Response

We once again thank the reviewer for his comments.

Changes are now tracked in yellow.

We added a phrase and a relevant citation regarding complications in the discussion section.

We once again think mentioning AF in the introduction is beyond the scope of our manuscript.

Furthermore, a visual representation of LVEF across studies is a great idea, but we cannot do it since some studies report LVEF gains in the whole population and others report LVEF gain only in the succesful ablation procedures.

Reviewer 4 Report

Comments and Suggestions for Authors

Congratulations to the authors for the revised version of their manuscript

Author Response

Thank you very much

Round 3

Reviewer 3 Report

Comments and Suggestions for Authors

The autors improved the manuscript. I have no others comments.